# Determining intrinsic potentials and validating optical binding forces between colloidal particles using optical tweezers

Chi Zhang [1], José Muñetón Díaz [1], Augustin Muster [1], Diego R. Abujetas [1], Luis S. Froufe-Pérez [1] & Frank Scheffold [1]

Understanding the interactions between small, submicrometer-sized colloidal particles is crucial for numerous scientific disciplines and technological applications. In this study, we employ optical tweezers as a powerful tool to investigate these interactions. We utilize a full image reconstruction technique to achieve high precision in characterizing particle pairs that enable nanometer-scale measurement of their positions. This approach captures intricate details and provides a comprehensive understanding of the spatial arrangement between particles, overcoming previous limitations in resolution. Moreover, our research demonstrates that properly accounting for optical binding forces to determine the intrinsic interaction potential is vital. We employ a discrete dipole approximation approach to calculate optical binding potentials and achieve a good agreement between the calculated and observed binding forces. We incorporate the findings from these simulations into the assessment of the intrinsic interaction potentials and validate our methodology by using short-range depletion attraction induced by micelles as an example.

Directly measuring interaction potentials between colloidal particles is essential for scientific studies and practical applications[1]. Understanding the nature and strength of particle interactions provides valuable insights into the behavior and properties of colloidal systems and enables advances in various scientific disciplines and technological fields. From a scientific perspective, studying the interaction potentials between colloidal particles contributes to our fundamental understanding of the forces and mechanisms that govern their assembly, aggregation, and self-organization[2]. It deciphers the intricate interplay of various physical and chemical factors such as electrostatics, van der Waals forces, steric effects, depletion forces, and solvent properties[3–5]. By accurately measuring these potentials, researchers gain insights into the stability, phase behavior, and structural transformations of colloidal systems, paving the way for developing theoretical and predictive models[6,7].

Beyond basic research, measuring interaction potentials has immense practical significance for applications in various fields.

Colloidal systems have wide-ranging applications in materials science, food, pharmaceuticals, biotechnology, and nanotechnology[8,9]. Understanding and controlling the interactions between colloidal particles is crucial for developing and optimizing the properties and functions of colloid-based materials, including advanced composites, drug delivery systems, photonic materials, and structural colors[10]. By accurately characterizing the interaction potentials, researchers can tailor the properties of colloidal systems to achieve desired outcomes, such as increased stability, controlled self-assembly, and tailored rheological behaviour[11]. Furthermore, the ability to measure and manipulate interaction potentials in colloidal systems has implications for developing new technologies. For example, colloid-based sensors, actuators, and microfluidic devices rely on precise control of inter-particle or particle-wall interactions to achieve desired functions and performance[12–14].

Scattering and microscopy techniques are generally unable to measure the interactions between colloidal particles directly.

[1]Department of Physics, University of Fribourg, 1700 Fribourg, Switzerland. ✉e-mail: chi.zhang2@unifr.ch; frank.scheffold@unifr.ch

However, they can provide insights into the structure and spatial correlations within large particle assemblies, allowing for extracting the pair correlation function and the potential of mean forces[15]. Notably, there is an exception regarding direct light scattering measurements of forces between paramagnetic colloids, where the anisotropy of dipole forces causes particles to form chains[16]. The Atomic Force Microscope (AFM) and the Surface Force Apparatus (SFA) offer a means to probe the forces between surfaces coated with colloidal particles directly or by attaching a colloid to the tip of an AFM[17,18]. Total internal reflection microscopy provides a high-precision method to measure the interaction potential between a colloidal particle and a wall[5,19].

One of the most versatile approaches for directly measuring particle-particle interaction potentials is through optical tweezers[20–22]. These techniques employ focused laser beams to trap and manipulate one or multiple colloidal particles. By tracking the positions of the trapped particles and analyzing their thermal fluctuations, one can infer the inter-particle forces and derive the corresponding interaction potentials. Optical tweezers offer flexibility and enable measurements over a wide range of conditions, making them a valuable tool in studying colloidal interactions[22–26].

Previously, the measurement of colloidal interaction potentials using optical tweezers was limited to center-to-center distances larger than the wavelength of light. This limitation was addressed by tuning long-range interaction potentials when studying small particles[24] or using particles with diameters larger than the wavelength[23]. However, both choices severely restrict the kind of colloidal systems that can be studied. Two main challenges existed in studying interactions between smaller particles at short distances. Firstly, accurately tracking small particles was challenging due to the overlap of their (diffraction-limited) images, making it difficult to localize or track the particle centers precisely; see also Fig. 1[27,28]. As a consequence, previous tweezer measurements encountered issues with tracking errors, which resulted in biases in certain experiments and raised doubts about the reliability of the outcomes[29]. Secondly, optical binding forces that arise from scattering become significant for particles smaller than the wavelength of light[30–32]. Although these forces are well-known, their exact modeling in the present framework has not been attempted so far[30]. We address both of these issues in the present work. Our method relies on full image reconstruction for particle tracking and quantitative modeling of optical binding forces using a discrete dipole approximation (DDA)[33]. It enables reliable and meaningful measurements of colloidal interaction potentials in previously unexplored parameter regimes by overcoming the challenges of precise particle tracking and accounting for optical binding forces.

## Results

### Assessing colloidal interactions with optical tweezers

We study common spherical colloidal polystyrene beads, radius $R$, measuring $2R = 500 \pm 20$ nm and $2R = 710 \pm 20$ nm in diameter. Our optical tweezer setup configuration employs a tightly focused near-infrared laser with a $\lambda = 1064$ nm wavelength. In optical tweezer configurations, such near-infrared lasers are frequently employed owing to their cost-effectiveness, minimal absorption in water, and the advantage of their wavelength not overlapping with the spectra of common fluorophores within the visible range. In our experiment the

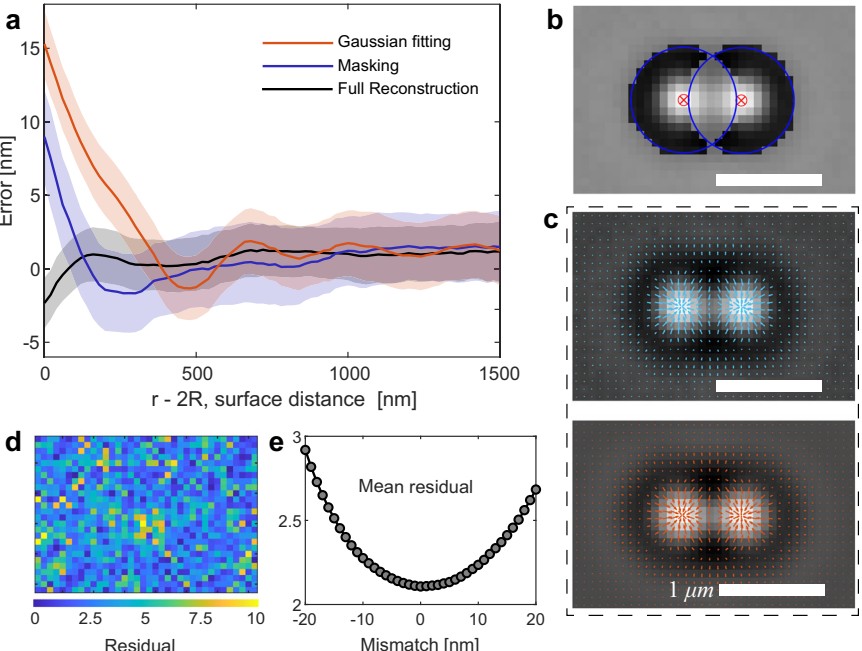

**Fig. 1 | Comparison of different particle tracking methods and their accuracy. a** Conventional single particle tracking algorithms utilize a particle-sized filter to blur the particle image, followed by fitting the blurred image with a Gaussian to determine the center position [22,56]. Image overlap introduces tracking errors when tracking multiple particles in close proximity leading to a systematic error in the determined position. The plot shows the tracking error as a function of the distance between a pair of particles, polystyrene beads of size (diameter) $2R = 500$ nm. **b** A masking approach, considering only the parts of the image that are non-overlapping, can be employed to improve the tracking accuracy [57–59]. While the masking approach outperforms the conventional method, it becomes ineffective for small particles where the residual area reduces to zero. **c** An alternative, model-independent approach for tracking two or more particles involves generating a synthetic image of the particle assembly using the individual particles' known image or shape function. The shape function can be acquired by imaging the particles before bringing them close together. The difference between the measured image (upper panel) and the reconstructed image (lower panel) is minimized by varying the particle positions. The arrows on the images indicate the corresponding intensity gradient. The full reconstruction method shows minimal bias across the entire distance range. The positional accuracy is about $\pm 2$ nm or $< 10^{-2}R$ close to contact, comparable to the statistical noise limiting the tracking precision (indicated by the shaded area). **d** The color-coded map displays pixel-by-pixel the residual (target error function, Eq. (3)), for the case with the best match. **e** The mean of the residuals as a function of the mismatch between a chosen and optimal position where the residuals become minimal. Scale bars are 1 μm.

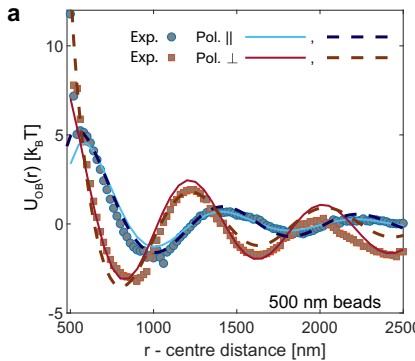
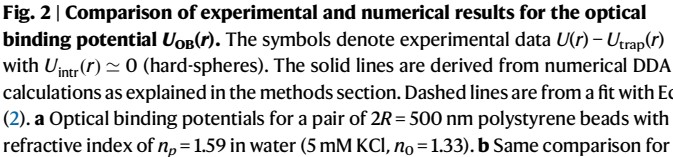
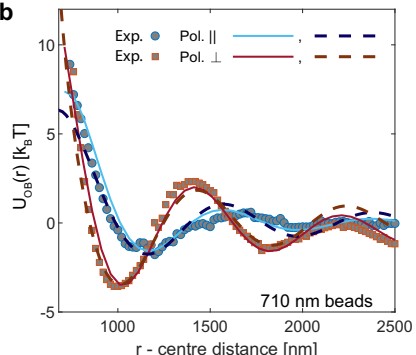

**Fig. 2 | Comparison of experimental and numerical results for the optical binding potential $U_{OB}(r)$.** The symbols denote experimental data $U(r) - U_{trap}(r)$ with $U_{intr}(r) \simeq 0$ (hard-spheres). The solid lines are derived from numerical DDA calculations as explained in the methods section. Dashed lines are from a fit with Eq. (2). **a** Optical binding potentials for a pair of $2R = 500$ nm polystyrene beads with a refractive index of $n_p = 1.59$ in water (5 mM KCl, $n_O = 1.33$). **b** Same comparison for a larger particle size of $2R = 710$ nm. The OB-potential is well-described by Eq. (2), where $r$ denotes the center-to-center distance between the particles, $\lambda$ represents the wavelength and $\alpha \simeq 0.8$. The $A$ and $\phi$ values obtained from the best fit are $[2k_B T, 1.3]$; $[3k_B T, 2.8]$ (left panel, ∥ and ⊥) and $[1.5k_B T, 0.1]$; $[2.25k_B T, 1.2]$ (right panel, ∥ and ⊥).

laser is directed into the sample cell using a holographic spatial light modulator (SLM) arrangement, previously described in [34,35]. Under typical experimental conditions, we estimate the laser power in the focal point to be about 100–150 milliwatts. We note that the laser power settings cannot be selected arbitrarily but are directly tied to the forces necessary for particle trapping. If the trapping force becomes too strong, it restricts the particle's Brownian motion, which, in turn, hinders our ability to investigate pairwise interactions. Consequently, the optical binding forces are also limited to a narrow range of possible adjustments.

To ascertain the positions of particles in close proximity, we utilize a comprehensive image reconstruction approach described in the methods section. Briefly, to obtain the image of a single particle, we trap and observe it, then determine its center position using the standard centroid tracking method[23]. Averaging over 1000 images allows us to obtain a low-noise image of the particle. To determine the precise center position of two particles nearby, we utilize numerical simulations to generate images corresponding to various center-to-center distances $r$. These simulated images are then compared to the experimentally captured images as shown in Fig. 1c. Minimizing the difference between images by varying the particle's position allows us to achieve optimal image reconstruction, enabling accurate determination of the particles' positions, Fig. 1e.

In thermal equilibrium, the probability of finding particles at a distance $r$ is related to the pair interaction potential $U(r)$ through a Boltzmann distribution: $P(r) \propto \exp[-U(r)/k_B T]$. This enables us to determine $U(r)/k_B T$ by analyzing the histogram of measured particle distances. The resulting interaction potential, $U(r)$, can be separated into three components:

$$U(r) = U_{intr}(r) + U_{trap}(r) + U_{OB}(r), \qquad (1)$$

where $U_{intr}(r)$ represents the intrinsic interaction potential of interest between the particles, $U_{trap}(r)$ corresponds to the well-known parabolic trapping potential, and $U_{OB}(r)$ describes the optical forces that emerge when the two particles are brought close to each other. Both $U_{trap}(r)$ and $U_{OB}(r)$ describe optical forces, but the distinction lies in the fact that $U_{trap}(r)$ can be determined by studying the individual particles and thus can be easily subtracted. On the other hand, $U_{OB}(r)$ only arises when the particles are in close proximity and cannot be eliminated using conventional approaches. Consequently, most experiments have been conducted using particles larger than the wavelength of the laser light, where optical binding forces are small and were thus ignored. It is important to note that extensive literature exists on observing and

modeling optical binding forces between tiny particles that otherwise interact as hard spheres[30,31]. In our work, we bridge the fields of optical binding and colloidal interparticle force measurements, allowing us to extract complex pair potentials for submicrometer-sized particles with interaction ranges spanning tens of nanometers.

We first examine polystyrene beads that exhibit repulsive interactions governed by a nearly hard-sphere potential. To achieve this, we suspend polystyrene beads in water containing an electrolyte concentration of 5 mM KCl, which results in a Debye screening length of $\lambda_D \simeq 4.3$ nm. Studying the interactions between particles behaving as hard spheres at greater distances enables the experimental determination of the optical binding potential, which we can then compare to numerical calculations. Figure 2 displays the experimental data for $U(r) - U_{trap}(r)$ obtained for two different particle sizes. It is important to note that the interaction potential is also influenced by the polarization of light in the optical tweezer. Using a retardation half-wave plate, we can adjust the polarization parallel or perpendicular to the long axis of the trap. Symbols of different colors represent the measured data sets for both cases. It is evident that the optical binding potential oscillates with an amplitude on the scale of several $k_B T$, and its binding energy gradually diminishes as the length scale increases to the micrometer level. Furthermore, selecting the polarization of the electric field of the tweezer to be parallel to its long axis reduces the effect, which is desirable.

## Quantifying optical binding potentials

To quantify the interaction energy of optical binding, we implement a full discrete dipole approximation (DDA) approach, as explained in the Methods section. The DDA predictions are depicted as solid lines in Fig. 2. We do not have exact knowledge of the power density of the incident light fields, so we adjust it to match the experimental data. Apart from this adjustment and a small level of uncertainty related to the particle radius, our calculations do not involve any tunable parameters and exhibit quantitative agreement with the experimental data as shown in Fig. 2.

In the presence of an additional potential of interest, with a finite range, the data at short distances, after subtracting $U_{trap}(r)$, will consist of a sum of $U_{intr}(r) + U_{OB}(r)$. Although the DDA approach demonstrates excellent performance, it would be advantageous to have a straightforward analytical expression for describing the optical binding potential at large $r-$values and then extrapolating it to smaller distances. In the subsequent text, we present and validate such an approach. The optical binding potential is expected to oscillate with a period determined by the wavelength ($\lambda = 800$ nm in water) and decay

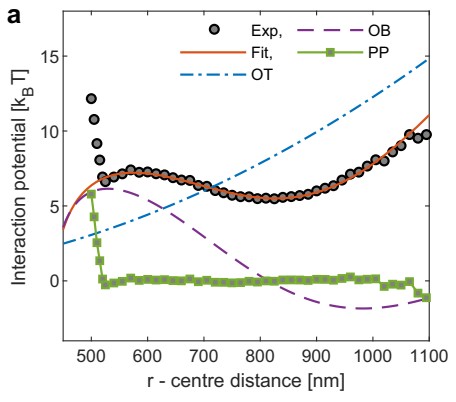
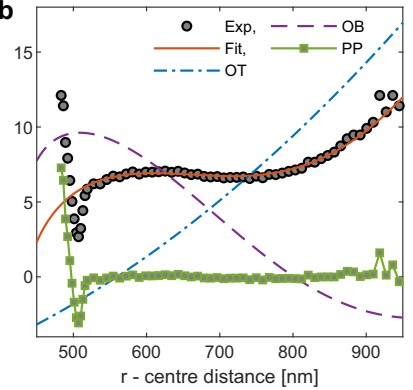

**Fig. 3 | Total interaction potential obtained from optical tweezer experiments in the absence and the presence of attraction.** Panel (**a**): Grey-filled circles denote experimental data for $U(r)$ obtained for polystyrene beads with a diameter of $2R = 500$ nm suspended in a water-based buffer (PBS 1X, $\lambda_D \approx 0.7$ nm)[46]. The blue dash-dotted line shows the optical tweezer (OT) harmonic potential $U_{\text{trap}}(r)$ determined experimentally. The red solid line shows the fit of the optical binding potential $U_{\text{trap}}(r) + U_{\text{OB}}(r)$ using Eq. (2) and $\alpha, \phi$-values taken from Fig. 2. $U_{\text{OB}}(r)$

from the fit is shown as a purple dashed line. Subtracting both contributions (OT +OB), we obtain the intrinsic particle-particle (PP) potential $U_{\text{intr}}(r)$ shown by green solid squares. These results reproduce the data shown in Fig. 2a) (∥ polarization). **b** Same analysis after adding 2 mM of the blockcopolymer pluronic F108 which are forming micelles and induce depletion attraction. The dash-dotted lines have been vertically offset for clarity. In both cases, the laser power settings differ, impacting the parameter 'A' in Eq. (2).

inversely with the distance between particles. One simple approach would be to use the Ansatz $U_{\text{OB}}(r) = A \cos\left[(2\pi r/\lambda + \phi)\right]/r$. However, it is known that this Ansatz fails when considering distances comparable to the particle diameter, $2R$. In light of this, we explored an alternative expression

$$U_{\text{OB}}(r) = A \cos\left[(2\pi r/\lambda + \phi)\right]/(r/2R - \alpha) \qquad (2)$$

The adjustable parameters for the fit are the amplitude, $A$, the phase, $\phi$ and an empirical parameter $\alpha$ of order one. We demonstrate the excellent agreement between the experimental data and our semi-empiric expression for $U_{\text{OB}}(r)$ in Fig. 2. From the fit, we derived a value of $\alpha$ approximately equal to 0.8, and $\phi$ values contingent on the size and polarization. The specific values are provided in the caption of Fig. 2. Consequently, throughout this study on polystyrene particles in water, we maintain these determined values constant.

### Accurate determination of intrinsic potentials
Having successfully modeled $U_{\text{OB}}(r)$, we investigate a system of interest exhibiting short-range attraction on the order of a few $k_B T$. To achieve this, we introduce the block-copolymer Pluronic F108 as a depletant. Under the specified conditions, F108 creates spherical micelles that induce attractive interactions between the polystyrene beads. In Fig. 3, we present experimental data for polystyrene particles with a diameter of 500 nm. The particles are suspended in a water-based buffer solution (PBS 1X) containing 2 mM pluronic F108 at a temperature of $T = 40°$. The polarization of the electric field of the tweezers is set to be parallel to its long axis to minimize the amplitude of $U_{\text{OB}}(r)$.

First, we subtract the trapping potential $U_{\text{trap}}(r)$. Subsequently, we fit the resulting potential at distances $r > 2R$, as illustrated in Fig. 3. Finally, we subtract the optical binding potential $U_{\text{OB}}(r) = A \cos\left[(2\pi r/\lambda + \phi)\right]/(r/2R - 0.8)$ to extract the intrinsic potential. The results depicted in Fig. 4, are obtained for two different particle sizes and two micellar concentrations above the critical micelle concentration (CMC).

Depletion forces arise in mixtures of large and small colloids, such as micelles or non-adsorbing polymers, where the small colloids act as depletants. The distance between the surfaces of the particles is denoted as $h$. Whenever $h$ is smaller than the diameter of the depletant micelle, the depletant particles are expelled from the region between the large spheres. Consequently, the concentration of depletant

particles becomes reduced in this region compared to the bulk, leading to an effective osmotic pressure that causes a net attraction between the large spheres. Asakura and Oosawa provided a quantitative explanation for this phenomenon[36]. According to their calculation, the change in the free energy can be expressed as $U(h) = -\frac{1}{2}\pi n R (d_m - h)^2$, in the case of $d_m \ll R$, where $n$ represents the number density of the depletant, $h$ is the surface-surface distance of the large particles, and $d_m$ is the diameter of the depletant micelle. The Asakura-Oosawa (AO) model considers the depletant (small) particles to behave like an ideal gas, and therefore, it is most accurate when applied to low to moderate particle densities. The interaction potential becomes oscillatory at larger depletant concentrations due to the liquid structuring[37,38], see Supplementary Material.

We observe a remarkable level of agreement between the measured potential $U_{\text{intr}}(r)$ and the theoretical model for depletion interactions among the polystyrene beads. The parameters of the Asakura-Osawa (AO) model consist of a micellar diameter of approximately 22 nanometers, as determined through dynamic light scattering, and an aggregation number of 45 polymers per micelle. This aggregation number aligns closely with the value reported in the literature, as indicated by reference [39]. To accommodate minor fluctuations in the size of the larger polystyrene beads, slight adjustments were necessary concerning the center-to-center distance between the particles $d$, as illustrated in Fig. 4.

## Discussion
In this study, we have employed optical tweezers as a powerful tool to investigate the interaction potential between submicron-sized colloidal particles with short-range interactions. Such interactions are common in model systems, practical applications, and natural systems. Therefore, it is crucial to characterize interactions between particles with subwavelength sizes, typically around 300–700 nm in diameter (or less), interacting on length scales of 20–40 nm. Previously, however, this specific configuration space has been largely unexplored due to limitations in available techniques, particularly optical tweezer technology. In this article we demonstrated that optical binding forces are important and must be considered when describing the total interaction potential and deriving the intrinsic potential, which refers to the potential in the absence of light fields. Using a discrete dipole model, we quantitatively describe binding forces for various particle sizes and incident wave polarizations. Based on these findings, we propose a straightforward analytical expression

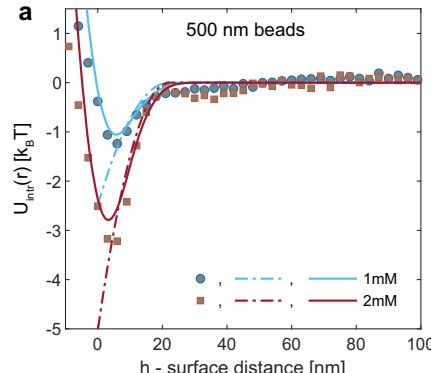
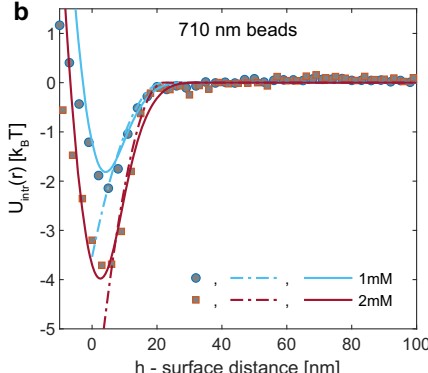

**Fig. 4 | Depletion interaction potential between polystyrene beads in Pluronic F108 (1 mM and 2 mM) water-based (PBS 1X) solutions ($T = 40\,^\circ$C). a, b** We show experimental results for $U(r)$ (squares and circles) for two different particle sizes, $2R \simeq 500$ nm and $2R \simeq 710$ nm. Every experimental dataset was obtained by selecting a distinct pair of particles. Dash-dotted lines show predictions by the Asakura-Oosawa model, calculated using $U(h) = -\frac{1}{2}\pi n R\,(d_m - h)^2$, where $n$ is the number density of the depletant micelles, $h$ is the surface-surface distance. $h = 0$

sets the contact distance between the particle's centers $d \simeq 2R$. We find a best fit for $d = 505$ nm (1 mM) and $d = 502$ nm (2 mM) in panel (**a**) and $d = 722$ nm (1 mM) and $d = 715$ nm (2 mM) in panel (**b**). $d_m = 22$ nm, the diameter of the depletant micelles, as obtained by dynamic light scattering (DLS). $n$ is calculated from the pluronic F108 concentration assuming an aggregation number of 45 for a best fit. Solid lines show the Asakura-Oosawa model prediction blurred by experimental errors, as described in the methods section.

to describe the binding potential. The few parameters in this analytical expression can be adjusted to match experimental data by fitting the potential at sufficiently large distances where the intrinsic potential becomes negligible. Through an example involving depletion forces induced by polymer micelles, we illustrate how accurately accounting for binding forces allows us to determine the intrinsic interaction potential.

The big advance we make with our approach is that it allows us to measure interaction potentials at small particle-particle distances, encompassing the case of particles with a small radius compared to the wavelength of light used for tracking and imaging. In contrast to conventional particle tracking algorithms, our approach does not impose a fundamental lower limit on the particle size or the surface-to-surface distance. The tracking approach and the DDA modeling could also be applied to plasmonic metallic or absorbing particles. Equally, the method can easily be extended to three or more particles as long as the computational cost of reconstructing images is not prohibitive. Studying three or more particles would allow us to assess the influence of multi-body interactions in dense particle assemblies[40]. The modeling of optical binding forces in our approach is limited to pairs of dielectric particles, but an extension to several particles or metallic particles is feasible, at least numerically, and should be addressed in future work.

While our tracking method delivers impressive positional accuracy, typically within the range of 2−3 nanometers, the center-to-center distance $d$ between two randomly selected particles in contact may exhibit a variation of 10−15 nanometers. This variation can be attributed to residual particle size differences, even when the size polydispersity is minimal, in our case just a few percent. Consequently, we were compelled to introduce minor adjustments to the center-to-center contact position (where '$h = 0$') in order to achieve a precise alignment between the measured interaction potential and the theoretical model, in scenarios involving short-range depletion interactions. While this introduces an additional degree of freedom into the fitting process, it's important to acknowledge that attaining higher precision would only be feasible if particle size measurements could be carried out 'in-situ' with an accuracy of 3 nanometers or better. Whether using, e.g. the approach of Bierbaum et al.[41] would allow achieving this particle sizing-precision under standard brightfield illumination and imaging is currently unknown and has to be addressed in future work. Despite the outstanding accuracy of

our tracking method, we still find a systematic deviation of 2−3 nm near contact and a statistical noise of comparable magnitude. While small, these finite deviations become visible in the model comparison, as shown in Fig. 4. Despite these residual challenges, our work substantially extends the range of particle sizes for which interaction potentials can be measured, covering the significant domain of submicron-sized particles. Our research will provide researchers with new tools to study interactions between functionalized colloids, opening up possibilities for research in colloidal sensing, diagnostics, and other applications[42−44].

## Methods

### Sample preparation

We studied two sizes of polystyrene (PS) beads (Bangs Labs, USA) with $500 \pm 20$ nm and $710 \pm 20$ nm diameter (supplier specifications), suspended in a water-based buffer solution. We either added 5 mM KCl to an aqueous suspension to screen electrostatic double-layer interactions, ensuring a nearly hard-sphere-like system due to the small Debye length ($\lambda_D \approx 4.3$ nm)[45]. Alternatively, we employed PBS 1X buffer for depletion measurements to maintain a pH of 7.4. In the PBS buffer, the Debye length is approximately $\lambda_D \approx 0.7$ nm[46]. We used a depletant system based on copolymer Pluronic F108 micelles to induce short-range attractive interactions and performed measurements at two different F108 concentrations. As shown in [39], the critical micelle concentration (CMC) of F108 is temperature-dependent, necessitating us to conduct our work at temperatures above the CMC value. We used dynamic light scattering (DLS) at a scattering angle of $90^\circ$ (3D Nanolab, LS Instruments Switzerland) to investigate the micelle formation and size of micelles at various temperatures. At $40\,^\circ$C, micelle formation was complete, and the hydrodynamic diameter of the micelles was measured to be $d_m = 22 \pm 1$ nm for both copolymer concentrations studied, namely 1 mM and 2 mM cases. Our findings are consistent with ref. 39. The latter estimated the micelle aggregation number between 35−61. For a value of 45 we obtain a number density of micelles $1.3 \times 10^4\,\mu\text{m}^{-3}$ and $2.6 \times 10^4\,\mu\text{m}^{-3}$ for 1 mM and 2 mM of F108, respectively. Consequently, the effective volume fraction of micelles $\phi = n \times \frac{\pi d_m^3}{6}$ correspond to 7% and 14%.

### Optical Tweezers experimental setup

We employ a Nikon Ti2 inverted microscope with an objective CFI Apo TIRF 100XC Oil. The numerical aperture (NA) of this objective is 1.49.

For optical tweezing, we use a $\lambda = 1064$ nm fiber laser (YLR-10-LP, IPG photonics). The laser beam first passes a spatial filter and is subsequently expanded and collimated to match the size of the Spatial Light Modulator's (X10468, Hamamatsu) active area (12 mm × 16 mm). After undergoing modulation with a computer-designed hologram, two Keplerian telescopes project the laser beam, which exits the Spatial Light Modulator (SLM), into the back aperture of the objective lens. The telescopes are organized in a $4f$ configuration, ensuring that the SLM chip and the back aperture of the objective align in conjugate planes. The hologram produced by the SLM consists of two components: the first part is dedicated to wavefront correction, employing the technique described in reference [47]. The second part is designed to generate a line trap, which can be either a Gaussian line or a combination of two truncated Gaussian lines, using the approach developed by Roichman and Grier[34].

We generate an optical potential as a Gaussian line to trap two particles simultaneously, enabling us to measure the pair potential between particles positioned close to each other. To ensure a sufficiently strong force that brings the particles together, we carefully adjust the stiffness of the trap[48]. We utilize two truncated Gaussian short-line traps to measure the pair potential at greater distances. This approach becomes necessary to maintain the particles at a desired average position and prevent them from getting trapped in local energy minima along the trap line. In both setups, we employ an sCMOS (scientific Complementary Metal-Oxide-Semiconductor) camera (Prime 95B, Teledyne Photometrics) to capture videos with an exposure time of 100 µs. Individual videos of single particles in each trap are recorded separately for the configuration using two truncated Gaussian traps. Using the particle tracking algorithm described below, we can extract distributions of the center-to-center distance (in the case of two particles) and the position distribution (for single particles). The latter distribution is subsequently utilized to calculate the potential of the optical trap, denoted as $U_{\text{trap}}$.

## Particle tracking algorithm

We employ a full image reconstruction method to determine the position of particles in close proximity. Prior image reconstruction algorithms have tried to assess particle size and have also been employed in fluorescent image analysis of micron-sized particles[41,49]. Each of these approaches presents distinct strengths and limitations. In our work, we propose an approach to precisely pinpoint the particle coordinates in brightfield imaging, particularly for particles with a size smaller or roughly equal to visible light wavelengths. The image of a single particle is obtained by trapping and observing it, and the center position is determined using the standard centroid tracking method[22]. By averaging over 1000 images, we obtain the particle's image, characterized by the shape function $S$. When dealing with spherical particles with a narrow size distribution, $S$ remains unchanged for all particles except for a size-dependent scaling factor. In the optical trap, a particle retains limited mobility along the $z$-direction, causing its $z$-position to deviate from the trapping plane during image acquisition. This results in a small change in the apparent size. Consequently, we can directly deduce $S$ for each particle from multi-particle images, assuming that the $S$ for individual particles can be derived by merely rescaling it to match their apparent size. Employing this procedure, our tracking algorithm is sped up significantly and can also be extended efficiently to assemblies of three or more particles. We have verified that the results are equivalent to measuring each particle's shape function. Since there are no interference patterns when using white light illumination, we can represent an image containing multiple particles as a sum of individual images $Reconstruction = \sum_i S(x_i, y_i, size_i, intensity_i)$, where $x_i$ and $y_i$ are the positions of the particles. $size_i$ and $intensity_i$ take into account the slight changes in apparent size and intensity. In practice, we utilize gradients in the $x$ and $y$ directions, represented as $G^x$ and $G^y$, to describe the image. Consequently, $S = [G^x, G^y]$, and the image is expressed as $Image = [G^x_{image}, G^y_{image}]$. This approach offers two advantages: the gradient is more sensitive in detecting shape edges and automatically excludes the background since its gradient is zero. With these gradient representations, we establish a target error function as:

$$Er = \left\| Image - Reconstruction \right\|^2$$
$$= \left\| \left[ G^x_{image} - \sum_i G^x(x_i, y_i, \ldots), G^y_{image} - \sum_i G^y(x_i, y_i, \ldots) \right] \right\|^2. \quad (3)$$

Since we're dealing with two particles, $i \in [1, 2]$, the target error function is a matrix with the same dimension as the image. By starting with an initial estimate of the coordinates obtained through the conventional centroid tracking method, the algorithm can effectively and robustly minimize Eq. (3) using a custom made coordinate descent algorithm[50]. We successfully attain the optimal image reconstruction, enabling us to determine the particles' positions accurately. Sample images and the MATLAB (The MathWorks Inc., USA) tracking code can be accessed from the online repository hosted on Zenodo. We note that generalizing the particle tracking algorithm by image reconstruction to three or more particles will be computationally more costly but otherwise straightforward.

## Experimental errors

There are two main contributors to the experimental error: localisation uncertainty associated static error and dynamic error due to finite camera exposure time [51]. In our study, tracking bias and noise were determined experimentally (see Supplementary Material). Briefly, the position of an immobile particle B (adsorbed on the coverslip) is monitored while a second particle C is brought close with optical tweezers. The coordinate system is calibrated using a fiduciary marker, particle A, to exclude the influence of possible drifts and vibrations. Any systematic change of the position of particle B upon approach of C must come from tracking bias. We find a tracking bias of 2–3 nm at contact, about an order of magnitude less than conventional approaches. The tracking noise can also be revealed from the statistics of the localisation of B and amounts to about ± 2 nm. The noise interval is depicted by the width of the gray shaded area in Fig. 1. In a potential measurement, we look at the distance between two particles. In this case, the statistical noise on distance (static error) is smaller than $2 \text{ nm} \times \sqrt{2} \simeq 2.8$ nm.

The dynamic error arises from the finite camera exposure time, leading to time-averaging of the particle's position in images. According to the Stokes-Einstein relation, a single particle with a size of 500 nm, observed over 100 µs, exhibits one-dimensional motion over distances of approximately 15 nm in the case of free diffusion. Consequently, the corresponding relative motion of two particles is around $\sqrt{2} \times 15 \text{ nm} \approx 20$ nm. When the separation between two particles falls below 20 nm (which is similar to the interaction range of our depletion potential), their relative motion will reduce to around 10 nm due to the proximity-induced decrease in the diffusion constant, as described in refs. 51,52.

These numbers are obtained with the assumption that two particles are both freely diffusive. In our case, however, the particles are confined due to depletion attraction which substantially reduces their relative motion. In an extreme case of infinitely strong attraction, the particles will be permanently bound. Therefore, their movement is fully coupled, resulting in zero relative motion. We can assess the relative motion by analysing its dependency on time delay and extrapolating the result to the exposure time of 100 µs as shown in the Supplementary Material. In our depletion measurements, we determine that the dynamic error remains below 5 nm (at separations, $h$, less than 20 nm) for all measurements.

The experimental errors blur the measurements of the interaction potential. Dynamic error leads to a time average over an interval of particle distances, and a static error adds a Gaussian noise to the measured distances. For the optical binding measurement (Fig. 2), these small errors are unimportant since this potential is fairly long-ranged and relatively smooth. For a short-ranged interaction potential due to depletion, the influence of blurring is non-negligible (Fig. 4). We model the influence of dynamic error by first converting the AO potential (together with the optical potentials) to a probability distribution $P(r) = \exp(-U(r)/k_B T)$. We consider the case that the particles explore a small distance range $r \pm \Delta$ ($\Delta \simeq 5$ nm), due to their relative motion within the exposure time. The probability of finding them exploring this distance range can be written as $Q(r) = \int_{r-\Delta}^{r+\Delta} P(a)\mathrm{d}a$. The variable $r$ is then replaced by the corresponding time-averaged distance $r'$, which can be described as the mean distance weighed by the probability distribution: $r' = [\int_{r-\Delta}^{r+\Delta} rP(r)\mathrm{d}r]/[\int_{r-\Delta}^{r+\Delta} P(r)\mathrm{d}r]$. Furthermore, we take account of the static error by convoluting $Q(r')$ with a Gaussian kernel with a half-width of 3 nm. Then we convert the final probability distribution back to obtain the blurred potential, as shown by the solid lines in Fig. 4.

## Computational analysis of optical binding forces

We numerically solve the optical binding problem by treating all light-matter interaction with the Discrete Dipole Approximation (DDA)[33]. In the DDA, scatterers are discretised in small cubes whose optical response is characterized by the polarizability of an equivalent point-dipole with the appropriate polarizability. Indeed, the approach is recognized for its ability to reach the precise solution[53], with its accuracy being solely constrained by limitations in time and memory. The latter limits the maximum number of dipoles to be used in the DDA.

We consider two identical spherical particles with diameters $2R = 500$ nm or $2R = 710$ nm and refractive index $n = 1.59$. The particles are placed in a background medium with refractive index $n_b = 1.33$. In both cases, particles are illuminated by an external input field, $\vec{E}_0$ modeling the linear trap, which is a beam with constant intensity profile along the $x$-axis, Gaussian intensity profile on $y$-axis and propagating along the $z$-axis.

Without loss of generality, the first particle (particle A) is centered at the origin of the coordinate system, while the other one, particle B, is placed at a surface-to-surface distance, $h$, on the $x$-axis. To calculate the optical forces induced by the external field, the optical response of each sphere is modeled in the DDA with $N = 1365$ small cubes with an edge length $D = (4\pi R^3/3N)^{\frac{1}{3}}$. The polarizability of each cube, $\alpha_i$, is given by the Clausius-Mossotti model with radiative corrections[54]. Thus, the total field, $\vec{E}_t$, exciting each dipole can be self-consistently calculated as a function of the input field. To evaluate the precision and convergence of our DDA simulations at reduced length scales, we compared results using a smaller number of dipoles, from 251 to 895 instead of 1365 dipoles per sphere. We examined the resultant forces at these levels of discretization. Our findings demonstrate that forces measured at distances less than 20 nm exhibit an accuracy exceeding 4%, extending down to a range of just a few nanometers.

The scattering problem can be represented as a system of $3N \times 2$ ($N$ dipoles per sphere) linear equations, where the three equations for the $i$-th dipole read:

$$\vec{E}_t^{(i)} = \vec{E}_0^{(i)} + k^2 \sum_{j \neq i}^{2N} \overleftrightarrow{G}^{(ij)} \alpha_j \vec{E}_t^{(j)}, \tag{4}$$

being $\vec{E}_t^{(i)}$ and $\vec{E}_0^{(i)}$ the total and external fields (resp.) at the $i$-th dipole, and $\overleftrightarrow{G}^{(ij)}$ the Green tensor that connects dipole $i$ and $j$. Equation (4) states that the total polarizing field at dipole $i$ is the sum of the external field plus the field scattered by the rest of the dipoles.

Once the total polarizing field at each dipole is known, the time-averaged force along the $x$-axis, $F_x(h)$, acting on particle B is computed using

$$F_x(h) = \frac{n_h^2 \epsilon_0}{2} \sum_{i=1}^N \left\langle \mathrm{Re}\left\{ \alpha_i \vec{E}_t^{(B_i)} \cdot \frac{\partial}{\partial x} \vec{E}_t^{(B_i)} \right\} \right\rangle, \tag{5}$$

where $\vec{E}_t^{(B_i)}$ stands for the total field on the $i$-th dipole of particle B. After computing the force in the inter-particle distance range $h \in [0, 60\,\mu m]$, the pair interaction potential is obtained by numerical integration of the force, $U(x) = \int_{60\mu m}^x F_x(h)\mathrm{d}h$, such that the force is $F_x = -\partial U/\partial x$.

The line trap is implemented by integrating its angular spectrum[55]. In the case of the electric field in x-polarization, the total input electric field reads

$$\vec{E}_{\text{x-pol}}(\mathbf{r}) = \int_{-k}^k \hat{E}_{\text{x-pol}}(k_y) e^{i(k_y y + k_z z)} \mathrm{d}k_y, \tag{6}$$

where $k_z = +\sqrt{k^2 - k_y^2}$, and

$$\hat{E}_{\text{x-pol}}(k_y) = \frac{E_0}{2\sqrt{\pi}} e^{-\frac{k_y^2 \omega_0^2}{4}} \mathbf{u}_x \tag{7}$$

Analogously, for the y-polarized line trap, we use

$$\vec{E}_{\text{y-pol}}(x,y,z) = \int_{-k}^k \hat{E}_{\text{y-pol}}(k_y) e^{i(k_y y + k_z z)} \mathrm{d}k_y \tag{8}$$

with

$$\hat{E}_{\text{y-pol}}(k_y) = \frac{E_0}{2\sqrt{\pi}} e^{-\frac{k_y^2 \omega_0^2}{4}} \frac{1}{k_z} \left[ k_z \mathbf{u}_y - k_y \mathbf{u}_z \right], \tag{9}$$

being $\mathbf{u}_i$ the unit vector along the $i-$axis. We notice that $k$ is the wave number in the host medium, and we use $w_0 = 200$ nm in the manuscript.

## Data availability

All experimental and numerical data generated in this study have been deposited in the the repository Zenodo under accession code 10245934 and 10245965. All additional data sets generated during and/or analysed during the current study are available from the corresponding author upon request.

## Code availability

Sample images and the MATLAB (The MathWorks Inc., USA) tracking code can be accessed from the same online repository hosted on Zenodo under accession code 10053674. The other codes used to produce the DDA-results of this study are based on a proprietary software library as described in detail in the manuscript.

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

## Acknowledgements

This research was financially supported by the Swiss National Science Foundation through the National Centre of Competence in Research (NCCR) "Bio-Inspired Materials," Grant No. 182881, along with Projects No. 149867 and No. 197146.

## Author contributions

C.Z. performed most of the experiments and data analysis. J.M.D. contributed to the optical tweezer experiments. A.M., D.R.A. and L.F.P. did the theoretical modeling of the optical binding potential and developed the DDA code. C.Z. designed the study and F.S. supervised the study. F.S. and C.Z. wrote the manuscript with contributions from all authors. All authors contributed to the analysis and interpretation of the data.

## Competing interests

The authors declare no competing interests.
