## [Peer Review File · Nature Communications]

Determining intrinsic potentials and validating optical binding forces between colloidal particles using optical tweezersREVIEWER COMMENTS

Reviewer #1 (Remarks to the Author):

In this paper, the authors measured the interaction between two colloids with diameters of 500 and 700 nm. They employed a conventional light microscopy technique to record the positions of the particles and proposed a new algorithm for extracting position information from the recorded images. The authors obtained the intrinsic interaction potential by subtracting the trapping potential and optical binding potentials from the gathered data. This paper introduces a new position detection technique for accurately measuring the interaction between two colloids, which is important for understanding micro and nano-scale interactions.

Before publication, I would like to see the answers to the following questions:

1. In the minimization of Equation 3, which type of searching algorithm did the authors use to obtain the parameters x , y , size, and intensity? This aspect should be explained in the manuscript.
2. In Equation 3, is "i" an index or a multiplier? It appears there might be a missing component here.
3. In Figure 1a, what does the label "BE" represent? Additionally, the scale bars in Figure 1b and c are not clearly comprehensible.

Reviewer #2 (Remarks to the Author):

In this work the authors have introduced an improved particle tracking scheme in order to investigate the interaction potential arising between two colloidal particles held at an optical line-trap. They have developed an intuitive particle tracking algorithm that compares an optical image with a numerically simulated image. By minimizing the difference between the recorded and simulated image a fully reconstructed synthetic image of a pair of colloids could be obtained. For a diffraction-limited image with over-lapping shape function this approach provides nearly accurate tracking of interparticle distances with low error and better than conventional centroid tracking method. Therefore, such tracking is utilized for

experimental observation of interaction potential with high spatial resolution. Further, they employed it to determine the optical binding potential and a short-range attractive potential arising from depletion interactions.

First, with colloids in hard sphere-type interaction the binding potential is plotted as a function of distance by subtracting the known trapping potential. The authors came up with a semi-empirical expression for binding potential which fits well with the experimental observations. The same expression is used to determine the intrinsic depletion interaction potential induced by micelles.

The work certainly provides a hierarchical approach and I think the manuscript could be ultimately suitable for Nature Communications. However, the following aspects, which I feel is necessary to consider for a paper of the level of Nat Comm:

- The technique relies on single particle image processing (with averaging over 1000 images) and then perform the reconstruction. Can such methods be implemented for multi-particle assemblies or dense suspension for that matter? What happens when polydispersity exists in the system? What is the accuracy for radii measurement?
- The current approach only considers model spherical particles. Can it be applicable to particles of non-spherical shapes? If so, please provide evidence.
- How does this method compare with relevant works which also can track particles extremely accurately such as Nature Communications 3, Article number: 1127 (2012); Phys. Rev. X 7, 041007
- The other concern is with the colloidal size range (500-1000 nm) where this approach can be adopted. I agree with the authors that for particles (500/700 nm) smaller than the trapping wavelength (1064 nm) will have significant binding force and I also appreciate that they have taken care of it. But one can argue that a blue laser may be used for trapping where the optical binding potential may be ignored. Also, the two decades old standard centroid tracking method (ref #13) uses particles of diameter 650 nm. Therefore, I feel this current approach becomes increasingly important if it can be extended for particles of sub-100 nm scale. Then it will also be interesting to a broad range of nanoparticles including plasmonic particles, several nanocrystals, fluorescent nanodiamonds etc. for which determining interaction potential should be of great interest. For example, decoupling depletion interactions from two optothermally trapped particles as demonstrated in Nature

Photonics 12, 195–201 (2018) or analyzing plasmonic trapping potential in Nature Communications 10, Article number: 4191 (2019) or for laser trapping of metal nanoparticles ACS Nano 2015, 9, 4, 3453–3469. This is also important considering the broad readership of Nature Communications.

- The authors should discuss the strengths and drawbacks of their methods in the form of a discussion.

Some minor comments:

- The citations are weirdly placed starting with Ref #37. This kills time while jumping around the citations.
- Please, stick to one form of notation, for example for particle size stick to either $2R$ or d .
- Add a citation for Debye screening length of 3 nm in 5 mM KCl solution.
- “Except the power density of the incident light fields our calculations lack any adjustable parameters.” Please recheck the sentence.

Reviewer #3 (Remarks to the Author):

The manuscript by Chi and coworkers describes line optical tweezers (LOT) measurements on sub-micron PS colloids, with an emphasis on understanding the separation dependent optical forces that a pair of spheres exert on one another. The measurements agree well with the result of DDA calculations, which is satisfying. To my knowledge, this work is new, and a nice contribution to the body of work measuring potentials using LOT.

I have concerns with the presentation regarding their particle tracking and analysis, in particular the micellar depletion part, which I think the authors will want to address.

--I felt the text "As a consequence, previous tweezer measurements encountered issues with tracking errors, which resulted in biases in certain experiments and raised doubts about the reliability of the outcomes (6)." was a bit overstated and lacking proper context. Many authors have published methods for correcting image centroids for image overlap for the last 20 years, but none are mentioned in this section. Bechinger's paper is correct, but

applies to >20 year old measurements, in particular by Grier, that did not correct for overlap at all.

--I was curious about the spatial resolution of contact separation h in their measurements. I get that maximizing the resolution is not their goal here (and not really critical for probing OB forces spanning 100s on nanometers of separation), but I think though that the authors could do some fitting of their micelle data, say to an Gaussian probability cloud, and reporting an upper bound for their resolution. Eyeballing it, it would seem to be a respectable <10 nm.

--Thinking further, the agreement between their AO model curve and their data appears fortuitous to me, since finite resolution would be expected to blur the potential, yielding a lower attractive force near contact. So either their model needs to be improved to account for resolution effects, or their extremely high resolution (so as to be negligible) needs to be demonstrated somehow.

--I was unclear whether their image analysis method found a center-center distance between two contacting particles that was equal to exactly $2R$, or if they needed to add some correction factor near contact as other authors do. If no correction was applied, the authors should state that clearly as a significant plus for their approach.

--There is also the matter of dynamic error. The authors use a shutter time of $1e-4$ seconds, and I think a 0.5 μm sphere diffuses about 14 nm in that time, or 20 nm for relative motion if uncorrelated. Since this is a greater distance than their apparent resolution above, it seems relevant. Likely the lubrication forces between the spheres reduce their relative motion to less than 20 nm, near contact. In practice, the associated blurring can alter the appearance of single beads slightly. I think such blurring bears a brief discussion.

--DDA is a great method, but I think the authors could say more about its limitations. It seems their model is less accurate near contact (not surprising given a finite dipole volume). Since many interactions concern the near contact region, I guess this is why the authors fit the DDA model to a sinusoid, and use the sinusoid instead of the DDA model. Is that

correct? This seems similar to earlier authors that fit optical forces near contact to empirical forms and subtract them off.

The authors might also consider citing: Biancaniello, Paul L., and John C. Crocker. "Line optical tweezers instrument for measuring nanoscale interactions and kinetics." *Review of scientific instruments* 77.11 (2006).

That paper considers many of the same tracking accuracy and precision issues as this manuscript, and also report measurements of micellar depletion in a LOT.

In summary, this is nice work measuring and modeling OB forces in a LOT, between well separated sub-micron particles. That part stands alone and is compelling in its current form. The part looking at micellar depletion is more problematic, not so novel, and requires a bit of additional work to understand the effects of resolution and dynamic error on the measurement before I am convinced it is correct and not fortuitous.

Reviewer #1 (Remarks to the Author)

In this paper, the authors measured the interaction between two colloids with diameters of 500 and 700 nm. They employed a conventional light microscopy technique to record the positions of the particles and proposed a new algorithm for extracting position information from the recorded images. The authors obtained the intrinsic interaction potential by subtracting the trapping potential and optical binding potentials from the gathered data. This paper introduces a new position detection technique for accurately measuring the interaction between two colloids, which is important for understanding micro and nano-scale interactions.

Before publication, I would like to see the answers to the following questions:

1. In the minimization of Equation 3, which type of searching algorithm did the authors use to obtain the parameters x , y , size, and intensity? This aspect should be explained in the manuscript.

In the minimization of Equation 3, we use the coordinate descent method to obtain the parameters. With an initial guess of the coordinates - which we obtain using the conventional centroid tracking method - minimization of Eq 3 can be achieved robustly using the algorithm.

We have added the above information to the manuscript.

“By starting with an initial estimate of the coordinates obtained through the conventional centroid tracking method, the algorithm can effectively and robustly minimize Eq. (3) using a custom made coordinate descent algorithm (49).”

2. In Equation 3, is " i " an index or a multiplier? It appears there might be a missing component here.

We thank the referee for noticing this point. We had a Latex syntax error in Eq. (3). " i " denotes the index for two or more particles. We have corrected this mistake.

3. In Figure 1a, what does the label "BE" represent? Additionally, the scale bars in Figure 1b and c are not clearly comprehensible.

The 'BE' was mistakenly added when preparing the Figure for publication. We have removed it. We added the $1\mu\text{m}$ scale bar label to all panels b,c to clarify the notation.

Reviewer #2 (Remarks to the Author)

In this work the authors have introduced an improved particle tracking scheme in order to investigate the interaction potential arising between two colloidal particles held at an optical line-trap. They have developed an intuitive particle tracking algorithm that compares an optical image with a numerically simulated image. By minimizing the difference between the recorded and simulated image a fully reconstructed synthetic image of a pair of colloids could be obtained. For a diffraction-limited image with over-lapping shape function this approach provides nearly accurate tracking of interparticle distances with low error and better than conventional centroid tracking method. Therefore, such tracking is utilized for experimental observation of interaction potential with high spatial resolution. Further, they employed it to determine the optical binding potential and a short-range attractive potential arising from depletion interactions. First, with colloids in hard sphere-type interaction the binding potential is plotted as a function of distance by subtracting the known trapping potential. The authors came up with a semi-empirical expression for binding potential which fits well with the experimental observations. The same expression is used to determine the intrinsic depletion interaction potential induced by micelles.

The work certainly provides a hierarchical approach and I think the manuscript could be ultimately suitable for Nature Communications. However, the following aspects, which I feel is necessary to consider for a paper of the level of Nat Comm:

- The technique relies on single particle image processing (with averaging over 1000 images) and then perform the reconstruction. Can such methods be implemented for multi-particle assemblies or dense suspension for that matter?

This method can be easily extended to multiple-particles. In fact, our algorithm is already multi-particle-ready. Thus, with modern computational resources, the method can easily be extended to several particles, although the computational cost quickly rises for large numbers of particles. Therefore, we believe the application to tens or hundreds of particles will be challenging.

We have added the following comment to the manuscript on page 16.

...our tracking algorithm [...] can also be extended efficiently to assemblies of three or more particles.

- What happens when polydispersity exists in the system? What is the accuracy for radii measurement?

Our method relies on the assumption of a single particle and azimuthal symmetry of the image (i.e. a sphere or a disk). In practice, we image one single particle and take an average over 1000 images. We obtain the particle general shape function (S in the main text) from this. Note that, in an optical trap, a particle can still move slightly in the z -direction (move away from the trapping plane). This leads to an “apparent size,” which is slightly different from the real size.

Our aim is not to determine the actual size accurately but to determine the particle center coordinate with utmost precision.

For rather monodisperse spherical particles (< 15%), one can obtain S for every particle directly from multi-particle images, assuming the S for individual particles can be obtained by simply rescaling S to match its apparent size which speeds up the algorithm significantly. For particles with large polydispersity, S might no longer be scalable (although we didn't try). In this case, a measurement of each particle's shape function can still be performed before they are brought close.

We have slightly rewritten the relevant section manuscript on page 16 to provide more detail about this aspect of our tracking algorithm.

- The current approach only considers model spherical particles. Can it be applicable to particles of non-spherical shapes? If so, please provide evidence.

We agree with the reviewer that applying the method to anisotropic particles would be interesting. The simplest case would be that the particle remains symmetric with respect to an axial direction, such as for ellipsoidal particles oriented parallel to the x,y -plane. A non-spherical shape adds a particle's in-plane orientation as an additional degree of freedom. Equation 3 then will be modified with new parameters accounting for the orientation. The current minimization method works only for a unimodal function. Whether a modified Equation 3 remains unimodal or not is uncertain, and we haven't explored this question. Despite the appeal of such a study, we think it would be well beyond the scope of our current work.

- How does this method compare with relevant works which also can track particles extremely accurately such as Nature Communications 3, Article number: 1127 (2012); Phys. Rev. X 7, 041007

The authors of Nature Communications 3, 1127 (2012) by Kurita et al. reported an interesting method of extracting the size of individual particles based on the 3D data of the coordinates of all the particles and for all the frames. Our algorithm has a different goal as it focuses on finding the center coordinates of particles. Our algorithm does not determine the size of the particles very accurately. In this sense, the two methods are qualitatively different.

The study described in Phys. Rev. X 7, 041007 (2017) by Bierbaum et al. introduced a tracking technique involving image reconstruction. This research primarily emphasized fluorescent particles and provided instances involving relatively large particles labeled with fluorescence, measuring 1.34 micrometers and 2 micrometers. The study conducted particle parameter fitting for (x, y, z, R) , taking into account factors like dye distribution and uneven lighting conditions.

In contrast, our approach centers on diffraction-limited brightfield images, particularly focusing on smaller particles. The primary objective of our algorithm is to handle images featuring

substantial overlaps of two or more particles. Consequently, it is clear that the two methods are fundamentally distinct in their approach.

Particularly in the context of tweezer measurements, intentional positioning of particles slightly outside the focal plane is common practice to enhance contrast in brightfield microscopy, resulting in an even greater degree of image overlap.

In summary, these two methodologies mentioned by the reviewer represent image reconstruction approaches. Nonetheless, they markedly differ from our approach in terms of qualitative aspects and possess distinct strengths and objectives.

We have added the following sentences to the manuscript (page 16).

Prior image reconstruction algorithms have tried to assess particle size and have also been employed in fluorescent image analysis of micron-sized particles~\cite{PhysRevX.7.041007, kurita2012measuring}. Each of these approaches presents distinct strengths and limitations. In our work, we propose an approach to precisely pinpoint the particle coordinates in brightfield imaging, particularly for particles with a size smaller or roughly equal to visible light wavelengths.

- The other concern is with the colloidal size range (500-1000 nm) where this approach can be adopted. I agree with the authors that for particles (500/700 nm) smaller than the trapping wavelength (1064 nm) will have significant binding force and I also appreciate that they have taken care of it. But one can argue that a blue laser may be used for trapping where the optical binding potential may be ignored. Also, the two decades old standard centroid tracking method (ref #13) uses particles of diameter 650 nm. Therefore, I feel this current approach becomes increasingly important if it can be extended for particles of sub-100 nm scale. Then it will also be interesting to a broad range of nanoparticles including plasmonic particles, several nanocrystals, fluorescent nanodiamonds etc. for which determining interaction potential should be of great interest. For example, decoupling depletion interactions from two optothermally trapped particles as demonstrated in Nature Photonics 12, 195–201 (2018) or analyzing plasmonic trapping potential in Nature Communications 10, Article number: 4191 (2019) or for laser trapping of metal nanoparticles ACS Nano 2015, 9, 4, 3453–3469. This is also important considering the broad readership of Nature Communications.

The standard centroid tracking method (Ref 13 by Crocker and Grier, PRL 1994, in the original manuscript) measured the pair potential of charge-stabilized particles with a diameter of 652 nm. However, their measurements were performed for center-to-center distances larger than ~1400 nm, as shown in their Figure 2. In this range (surface distance > 700 nm), as emphasized in our work and shown in our Figure 1 a, the tracking bias is small, irrespective of the particle's size. Our method shows its advantages only when the trapped particles are close to contact (which is the more common case).

We agree that some measurements using a visible wavelength laser will indeed result in a simpler OB potential, reducing the problem discussed in our work. The problem, however, will

just be shifted towards a smaller particle size, and our solution still applies then. As emphasized correctly by the referee, we could then successfully target particles in the 100nm range. For practical reasons and for a proof of principle, we have chosen to work using common settings, but our study clearly provides the groundwork to go much further in this direction of small particle sizes.

Moreover, for many users of particle tweezing experiments, we argue that phototoxicity and stimulating fluorescent emission using blue light often creates problems, which is the reason (together with the low cost) most researchers work with a 1064nm laser for particle trapping.

- The authors should discuss the strengths and drawbacks of their methods in the form of a discussion.

We have incorporated this discussion within the 'Discussion' section.

Some minor comments:

- The citations are weirdly placed starting with Ref #37. This kills time while jumping around the citations.

We apologize for this mishap due to a Latex formatting problem. We have corrected the error in the revised version of the manuscript, and the citations are now ordered correctly.

- Please, stick to one form of notation, for example for particle size stick to either $2R$ or d .

We now use R ; if we speak of the diameter, we use $2R$.

- Add a citation for Debye screening length of 3 nm in 5 mM KCl solution.

We have added the citation: A. M. Smith, A. A. Lee, and S. Perkin. The electrostatic screening length in concentrated electrolytes increases with concentration. *The Journal of Physical Chemistry Letters*, 7(12):2157–2163, 2016.

In the introduction, the authors of this work state explicitly that a 1mM monovalent salt solution (such as KCl or NaCl) has a Debye screening length of 9.6nm. Since the screening length scales with the inverse of the square root of the ionic strength (as stated on the same page of this reference), a 5mM solution has a Debye screen length of 4.3nm. We have corrected our initial value for the screening length and added the reference.

- “Except the power density of the incident light fields our calculations lack any adjustable parameters.” Please recheck the sentence.

We replaced this sentence by the following corrected version on page 7:

We do not have exact knowledge of the power density of the incident light fields, so we adjust it to match the experimental data closely. Apart from this adjustment and a small level of uncertainty related to the particle radius, our calculations do not involve any tunable parameters and exhibit quantitative agreement with the experimental data as shown in Figure 2.

Reviewer #3 (Remarks to the Author)

The manuscript by Chi and coworkers describes line optical tweezers (LOT) measurements on sub-micron PS colloids, with an emphasis on understanding the separation dependent optical forces that a pair of spheres exert on one another. The measurements agree well with the result of DDA calculations, which is satisfying. To my knowledge, this work is new, and a nice contribution to the body of work measuring potentials using LOT.

I have concerns with the presentation regarding their particle tracking and analysis, in particular the micellar depletion part, which I think the authors will want to address.

--I felt the text "As a consequence, previous tweezer measurements encountered issues with tracking errors, which resulted in biases in certain experiments and raised doubts about the reliability of the outcomes (6)." was a bit overstated and lacking proper context. Many authors have published methods for correcting image centroids for image overlap for the last 20 years, but none are mentioned in this section. Bechinger's paper is correct, but applies to >20 year old measurements, in particular by Grier, that did not correct for overlap at all.

We would like to thank the reviewer for these remarks. We have added a number of more recent references as follows.

Bierbaum, M., Leahy, B.D., Alemi, A.A., Cohen, I. and Sethna, J.P., 2017. Light microscopy at maximal precision. *Physical Review X*, 7(4), p.041007.

Parthasarathy, R., 2012. Rapid, accurate particle tracking by calculation of radial symmetry centers. *Nature methods*, 9(7), pp.724-726.

Yücel, H. and Okumuşoğlu, N.T., 2017. A new tracking algorithm for multiple colloidal particles close to contact. *Journal of Physics: Condensed Matter*, 29(46), p.465101

Cheong, F.C., Krishnatreya, B.J. and Grier, D.G., 2010. Strategies for three-dimensional particle tracking with holographic video microscopy. *Optics express*, 18(13), pp.13563-13573.

van der Wel, C. and Kraft, D.J., 2016. Automated tracking of colloidal clusters with sub-pixel accuracy and precision. *Journal of Physics: Condensed Matter*, 29(4), p.044001.

Burov, S., Figliozzi, P., Lin, B., Rice, S.A., Scherer, N.F. and Dinner, A.R., 2017. Single-pixel interior filling function approach for detecting and correcting errors in particle tracking. *Proceedings of the National Academy of Sciences*, 114(2), pp.221-226.

-- I was curious about the spatial resolution of contact separation h in their measurements. I get that maximizing the resolution is not their goal here (and not really critical for probing OB forces spanning 100s on nanometers of separation), but I think though that the authors could do some fitting of their micelle data, say to an Gaussian probability cloud, and reporting an upper bound for their resolution. Eyeballing it, it would seem to be a respectable <10 nm.

We have integrated a discussion on tracking precision and resolution into both the main text and the supplementary information for more comprehensive coverage.

--Thinking further, the agreement between their AO model curve and their data appears fortuitous to me, since finite resolution would be expected to blur the potential, yielding a lower attractive force near contact. So either their model needs to be improved to account for resolution effects, or their extremely high resolution (so as to be negligible) needs to be demonstrated somehow.

We thank the reviewer for this comment which gave us the opportunity to improve our model comparison. While the agreement with the basic AO-model remains valid, the reviewer's observation regarding our oversight of the blurring effect resulting from the finite resolution of our methods is well-founded. In the revised version of the manuscript, we have addressed this concern by incorporating a detailed discussion and by including the corresponding plots in the revised Figure 4.

Additionally, we've recognized that our use of the term "parameter-free" might be overstated. We removed the mention 'parameter-free'. In our previous graph, we selected a micelle aggregation number of 60 to match the observed interaction strength. Considering the impact of errors, our updated plots now employ a micelle aggregation number of 45. Nevertheless, it's worth noting that both 60 and 45 fall within the range of aggregation numbers reported in the literature, reference (44).

--I was unclear whether their image analysis method found a center-center distance between two contacting particles that was equal to exactly $2R$, or if they needed to add some correction factor near contact as other authors do. If no correction was applied, the authors should state that clearly as a significant plus for their approach.

The center-center distance between two contacting particles found by our tracking method is almost exactly $2R$, with a very small correction of a few nm. In Figures 2 and 3, we needed no correction for the contact position because these plots are not sensitive to the difference of a few nanometers. But in Figure 4, we have added very minor corrections. For instance, $h = 0$ corresponds to 505 nm in the left panel (1 mM) (for a nominal particle diameter of 500nm).

In our experiments we use commercial polystyrene beads with a nominal size provided by the manufacturer, probably based on electron microscopy or dynamic light scattering. However, since we consider individual particles taken from an ensemble with finite polydispersity, already the standard deviation of about 3% would mean that our population of particle sizes is distributed over a range from 485nm to 515nm.

--There is also the matter of dynamic error. The authors use a shutter time of $1e-4$ seconds, and I think a 0.5 μm sphere diffuses about 14 nm in that time, or 20 nm for relative motion if uncorrelated. Since this is a greater distance than their apparent resolution above, it seems relevant. Likely the lubrication forces between the spheres reduce their relative motion to less

than 20 nm, near contact. In practice, the associated blurring can alter the appearance of single beads slightly. I think such blurring bears a brief discussion.

In our particular case, the dynamic error is small but indeed it contributes to blurring. Recognizing the complexity of addressing the dynamic error issue, we have added a summary of the experimental error treatment to the methods section of the main manuscript. Moreover, we added supplementary information that provides a detailed discussion of this matter.

--DDA is a great method, but I think the authors could say more about its limitations. It seems their model is less accurate near contact (not surprising given a finite dipole volume). Since many interactions concern the near contact region, I guess this is why the authors fit the DDA model to a sinusoid, and use the sinusoid instead of the DDA model. Is that correct? This seems similar to earlier authors that fit optical forces near contact to empirical forms and subtract them off.

The referee's observation is accurate, and it's true that the Discrete Dipole Approximation (DDA) exhibits reduced accuracy on small length scales. In order to assess the accuracy and convergence of our DDA simulations at smaller length scales, we expanded our DDA simulations in two ways. Firstly, we increased the number of dipoles, with up to 1365 dipoles per sphere (compared to 596 in the original analysis), and secondly, we compared the resulting forces at different levels of discretization. Our findings indicate that forces at distances below 20nm exhibit precision better than 4% down to a few nanometers.

Finally we note that the fit of the DDA model to a sinusoid is done mainly to make our method for correction for optical binding forces accessible to other researchers, although it is also useful for generalizing the result beyond the DDA accuracy.

The authors might also consider citing: Biancaniello, Paul L., and John C. Crocker. "Line optical tweezers instrument for measuring nanoscale interactions and kinetics." *Review of Scientific Instruments* 77.11 (2006).

That paper considers many of the same tracking accuracy and precision issues as this manuscript, and also report measurements of micellar depletion in a LOT.

We have added the citation and thank the reviewer for pointing it out to us.

In summary, this is nice work measuring and modeling OB forces in a LOT, between well separated sub-micron particles. That part stands alone and is compelling in its current form. The part looking at micellar depletion is more problematic, not so novel, and requires a bit of additional work to understand the effects of resolution and dynamic error on the measurement before I am convinced it is correct and not fortuitous.

We would like to stress that we did the micellar depletion measurements, as an important and challenging example, to demonstrate the power and accuracy of our approach.

Still, we appreciate the comments concerning the ‘resolution and dynamic error on the measurement’ which we have now addressed in the revised version of the manuscript. This addition substantially improves the quality of our paper.

REVIEWERS' COMMENTS

Reviewer #1 (Remarks to the Author):

The revised manuscript is now entirely acceptable. The authors have diligently addressed all of my comments. The shared code on Zenodo is well documented. Congratulations to authors.

Reviewer #2 (Remarks to the Author):

The authors have sufficiently addressed my comments. I recommend the publication of the article in Nat Comm. An additional suggestion, in relation to the potential applicability of the technique to smaller nanoparticles as mentioned in the revised version, they might consider citing a few relevant references (Nature Photonics 12, 195–201 (2018); Nature Communications 10, Article number: 4191 (2019); ACS Nano 2015, 9, 4, 3453–3469 etc.) in the context of nanooptical trapping where this technique can be interesting, thus highlighting the broad scope of the work.

Reviewer #3 (Remarks to the Author):

I have no concerns regarding the revised manuscript.

I was pleased to see that the authors improved the fitting procedure of their micellar depletion data, and better described the literature context for their work.